# Experiences with Testing, Self-Isolation and Vaccination in North East England during the COVID Pandemic

**DOI:** 10.3390/vaccines9070759

**Published:** 2021-07-07

**Authors:** Richard Harris

**Affiliations:** Department of Economics & Finance, Durham University, Durham DH1 3LB, UK; r.i.d.harris@durham.ac.uk

**Keywords:** vaccine hesitancy, COVID-19, ethnicity, multivariate regression

## Abstract

This study was based on a (population weighted) sample of some 4533 responses to a household survey conducted in March 2021 that looked at the impact of COVID-19 on residents in most of the local authorities covering the North East of England. It considered the outcomes relating to needing a COVID test, self-isolating, whether residents agreed that UK government and NHS-approved vaccines were ‘very safe’, and whether they had enough information in order to make an informed decision about whether or not to get vaccinated. Modelling these outcomes using multivariate regression produced a range of results that showed that all of the following were important: the impact of age, living in deprived areas, ethnicity, religious affiliation, disability, industry, occupation, economic status, changes in household income, sexual orientation, and household composition. Thus, the results showed that there are complex socioeconomic factors associated with the willingness to get a test, self-isolate, and the levels of vaccine hesitancy, such that, in future ensuring that (re-)vaccination and ‘track and trace’ programmes are successful, may need to be better nuanced by references to such factors rather than adopting programmes that mostly just rely on age as the criteria for roll-outs.

## 1. Introduction

The COVID-19 pandemic (hereafter, C19) has had an unprecedented impact on people and economies since it began to spread globally at the beginning of 2020. It is generally accepted now in mid-2021 that the optimal way to tackle C19 in terms of mitigating its socioeconomic and health impact is through (i) establishing ‘herd immunity’ that will significantly reduce the spread of the disease, and this principally means ensuring somewhere between 67% and 80% of the population of the UK is (re-)vaccinated, and (ii) ensuring an effective containment of outbreaks (especially any new variants) through ‘track-and-trace’, and this is reliant on people being willing and able to both test for the disease and then self-isolate if testing produces a positive result.

There has been considerable discussion of the extent to which vaccine uptake is lower in more deprived areas [1,2] and, especially, whether those from non-White ethnic groups place less trust in the health system more generally and are therefore less likely to engage with, especially, the vaccination programme. Specifically, a recent study [3] surveyed 9390 respondents in late November 2020 to statistically identify those mostly likely to exhibit vaccine hesitancy [4], along with the reasons for such hesitancy. Noting that respondents were asked for information ex ante, since vaccine roll-out did not begin in the UK until early December 2020, overall, for the UK, some 53.5% of participants stated they were very likely to be vaccinated, with a further 28.5% saying they were likely, leaving 18% classified as vaccine-hesitant. When disaggregated into subgroups, the study showed higher vaccine hesitancy for females (log odds of 1.68 higher compared to males); younger people (i.e., 1.64 higher for those aged 25–34 vs. 45–54 year olds, the reference group); and certain ethnic groups (led by Black/Black British with a 12.96 higher ratio, then Pakistani/Bangladeshi at 2.31, followed by mixed ethnicity at 2.24). The main reasons for vaccine hesitancy were concerns over unknown future effects (42.7% stating this as the main reason), although for Black/Black British, the main reasons were unknown future effects (30%) and a lack of trust in vaccines (29%); for Pakistani/Bangladeshi, the most important reasons were concerns about side effects (36%) and unknown future effects (35%). Ex post NHS England data discussed in reference [5], based on reference [6], showed substantially lower rates of vaccinations among those over 80 in ethnic minority subgroups and deprived communities, e.g., between 8 December 2020 and 17 March 2021, 94.7% of patients aged ≥ 80 not in a care home received a vaccine (with substantial variations such as: White, 96.2% vaccinated and Black, 68.3% and least-deprived, 96.6% and most-deprived, 90.7%).

Other research [7] has shown that non-White ethnic groups have experienced higher infection rates from C19, hospitalisation, and death, and this is explained by (inter alia) their being “… more likely to live in crowded and multi-generational households where self-isolation and social distancing may prove to be difficult… individuals living in deprived areas have higher diagnosis and death rates… (and) social distancing was effective and possible in higher socioeconomic level households” (p. 1). It was also noted that ethnic minorities were also more likely to work in certain industries with a higher risk of exposure, such as food retail, health and social care, and transport. These groups experience a lower uptake of vaccines because of a lack of trust resulting from prior “… cultural and structural racism, low confidence in the safety and efficacy of the vaccine… moreover, physical barriers including lack of vaccines, transport access and inconvenient appointments can also hinder vaccine uptake in these communities” ([8], p. 2).

A major survey undertaken to understand vaccine hesitancy is the Oxford Coronavirus Explanations, Attitudes, and Narrative Survey [9], which obtained responses from 5114 UK adults between 24 September and 17 October 2020. It found some 28.3% of the population could be labelled as vaccine-hesitant. The major task was to explain the reasons for this hesitancy, finding that the major reasons were that respondents thought vaccine data are fabricated (20% of the sample), while 25% did not know whether such fraud is occurring or not. Importantly, the study found that mistrust was evident across the entire population and only “… slightly higher in young people, women, those on lower income, and people of Black ethnicity” ([10], p. 2). In contrast, reference [11] found that during weeks 9–12 of the first national lockdown (May to June 2020) some 26% of Scottish participants (based on a sample of 3436) could be grouped as vaccine-hesitant (by August 2020, this fell to 22.5% for the 2016 respondents who stated they remained hesitant when completing a follow-up survey). Based on a multivariate analysis of the pooled samples that included age, ethnicity, education, household income, and those at high risk/shielding, the study found that gender and age were not statistically significant as a predictor of vaccine uptake, but those of White ethnicity were almost three times as likely to get vaccinated as Black, Asian, and minority ethnic (BAME) groups (high income and highest education subgroups were also more likely to accept future vaccinations, as were those shielding).

Lastly, polling data in December 2020 [12] found the vaccine hesitancy in the UK population at around 24%, although this rose to 43% for those from ethnic minority backgrounds and 30% for low-income earners (women had a slightly higher level of hesitancy compared to the overall population at 27%). Data reported in reference [13] in February also showed that, when comparing the NHS vaccination data with Public Health England’s deprivation scores, “… that six of the most deprived areas in England were in the bottom 10 local areas for vaccine uptake among the over-80 s and those aged over 75”. Based on the same data sources, reference [14] reported on similar differences across rich and poorer localities.

This paper makes a contribution by looking at the extent to which a wide range of personal characteristics and locations are associated with the C19 questions asked in the North East Covid Survey undertaken in March 2021 to provide information relevant to how to run the most effective vaccination and ‘track-and-trace’ programmes.

## 2. Materials and Methods

An online survey was conducted by the Gateshead Council from 8 to 28 March 2021 resulting in 5556 responses and a sample size of 4533 available for a subsequent analysis (when nonresponses to key variables like age, ethnicity, economic status, and gender are taken into account). It included 8 other local authorities in the North East administrative region of England (see Figure SA.1 in a longer version of this paper [15]), and jointly through their Directors of Public Health, they promoted the survey via various (resident) groups and other email lists available to these councils. In addition, all employers with 50 or more employees operating in the area were identified using the Orbis company database, and these were contacted and asked to promote the survey to their employees. The first question in the survey itself asked which local authority did the respondent live in, and, together with a question on their postcode, this was used to filter out a very small number of noneligible returns. Data from the Quarterly Labour Survey conducted by the ONS [16] was used to construct weights to ensure the survey was representative of the underlying population. Details on the distribution of respondents by the local authority, and the other subgroupings of the data used to construct weights, is available in Appendix 2 of reference [15].

The focus of the survey was the impact of C19 on how it affected people, their families, employment situations, and incomes, as well as views about a post-pandemic future. However, the focus in this paper was on a set of questions relevant to the NHS obtained from reference [17] that were also included during testing for C19, self-isolation, and vaccines; these questions are reproduced in Appendix A. A range of questions on personal and household characteristics (such as age, gender, ethnicity, religious affiliation, disability, sexual orientation, household composition, economic status, industry and occupation subgroups, and location) were included to consider the extent to which the outcomes were correlated with these characteristics. Table 1 provides the (weighted) means and standard deviations for these ‘explanatory’ variables, noting that cross-sectional modelling as undertaken here cannot establish causal relationships but only correlations.

Table 1 shows that the analysis reported in the next section was based on residents with an average age of just over 52 years old, some 48% were male, only 3.1% were of non-White ethnicity, and there was a very wide range in deprivation levels across the areas in which people lived.

Some 30% of the respondents stated C19 had a negative impact on the household finances, and 24% stated that the pandemic resulted in a change in their employment status. As to religious affiliation, the largest subgroup reported as being Christian (some 53%) followed by ‘no faith/religion’ (44%). Some 27% stated they had a disability (mostly limiting mobility ‘a little’), and the most important industry sector was human health and social work activities, followed by other services. Some 27% of respondents stated they were part of the ‘professional’ socioeconomic subgroup, with administration and secretarial occupations the next most important. Some 7% of respondents reported as non-heterosexual. The average number of adults in a household was close to 2, and some 37% reported children in the household. The dominant household composition was couples (some 41% with no children and a further 26% with children), followed by around 18% of households comprising single occupants. Durham County had the largest share of residents (26%), with Middlesbrough and Darlington both accounting for about 5% of the 16+ resident population.

## 3. Results

A longer version of this paper [15] contains a set of results based on univariate analyses that showed that there were statistically significant differences, in terms of the impact of C19, across various subgroups (e.g., ethnicity, age, whether there were children in the household; see Section 3 of reference [15]). However, as Table 2 shows, many of these differences do not remain statistically significant when other covariates (cf. Table 1) are introduced into multivariate models that seek to determine which factors have the strongest associations with the outcomes.

The outcomes that were subjected to multivariate testing considered which covariates were more strongly associated with:The 55% of residents who needed a COVID test;The 47% of residents who needed to self-isolate;The just over 55% (by March 2021) who had received at least one dose of a vaccine;The 18% who stated they faced a challenge getting a test;The nearly 31% who faced challenges when self-isolating;The 48% (57%) stating they strongly agreed that the UK government (NHS)-approved vaccine was very safe; andThe 51% who stated they strongly agreed they could make an informed decision about being vaccinated or not.

It is worth noting that, with respect to whether residents disagreed (or were neutral) about the statements underlying the last two bullet points, overall, the percentage who disagreed (including and/or excluding those who neither agreed or disagreed) with these three statements was relatively small; including those who disagreed and/or were neutral, it was 17%, 13.5%, and 19.5% for the UK government-approved vaccine, NHS-approved vaccine, and the ability to make an informed decision. Excluding the neutral answers, these percentages fall to 4.3%, 2.9%, and 6.2%, respectively. This suggests that, when compared to the ex ante information reported in early studies using the pre-roll-out 2020 data, vaccine hesitancy may have substantially declined (assuming the North East is representative of other areas and the UK as a whole).

A multivariate regression analysis was undertaken; and for dependent variables that were dichotomous (no/yes coded as 0/1), (weighted) probit regression was used. When the dependent variable (e.g., whether the UK government-approved vaccine is very safe) had more than a 0/1 outcome, ordered probit regression was used. The results obtained are provided in Table 2; note, for the ordered probit models, only the results for the largest subgroup (i.e., those who strongly agreed) are reported (full results are provided in reference [15], Tables SA2.9–11). Marginal effects are provided; for discrete (0/1) explanatory variables (cf. Table 1), these indicate the increase in the probability of the outcome (e.g., needed testing) from switching someone from 0 to 1 (e.g., moving from a non-White to White ethnic status). For continuous variables (age and the index of multiple deprivation), the marginal effect shows the increase in the probability of the outcome for a unit change in the explanatory variable (e.g., the effect of increase from being 25 to 26 years old).

To aid interpretation, the first column of results in Table 2 are presented variable-by-variable: as the age of the respondent increases, the need for testing declines. As shown diagrammatically in Figure S3 in reference [15], for those aged 20 years, the probability of needing a test is 0.66—or 66%—and this declines to 0.46—or 46%—for those aged 80 years. Thus, the marginal effect of moving from 20 to 80 years is a ceteris paribus (cet. par.) decline in the probability of needing a test of 0.20 (or 20%). Those belonging to the White ethnic subgroup were some 12.6% less likely (vs. other ethnic groups) to need a test. Those of the Buddhist faith had a much (nearly 40%) higher probability of needing to test, while, for Christians, there was a 6.8% higher probability (compared to those with no faith/religion). Having a major disability increased the (cet. par.) need for testing by over 13% compared to those without disabilities. Those working in the mining and quarrying and transportation sectors were more likely to need testing (47% and nearly 29% more likely, respectively), while those working in wholesale distribution were nearly 31% less likely to need testing. The skilled trades occupation subgroup was associated with around a 10% less need for testing, and those who experienced a change in their employment status during the pandemic were over 8% more likely to need testing. Having children in the household increased (cet. par.) by over 15% the likelihood of a need to test for C19, while households with only single-person occupancies were some 6% less likely to need testing. Lastly, those resident in Darlington, Middlesbrough, and Redcar and Cleveland (areas where the transmission rates were known to be higher) were between 9% and 17% more likely to need a COVID test.

Rather than go through the results in Table 2 column by column, the alternative used here is to summarise the impact of each of the determinants on the range of outcomes considered. Starting with age, this was significantly and negatively related to needing a test, challenges to testing, to isolating, and to getting a vaccine, and increases in age was positively associated with having had at least one vaccination dose by March 2021. Moving from someone aged 20–80 years reduced (the probability of) the need for testing, facing a challenge getting a test, and isolating or getting a vaccine by 20%, 9%, 13%, and 27%, respectively. In contrast, the likelihood of receiving a vaccine increased by 78% over this age range. This was after having controlled for family characteristics, religious affiliation, sexual orientation, ethnicity, and other socioeconomic characteristics, including where people lived. There was no statistically significant age effect on being able to make an informed decision on whether to vaccinate, and overall, the results presented here showed a much smaller level of vaccine-hesitant attitudes associated with age when compared to reference [3]; instead, the results were more in line with those found in reference [9]. Still, younger people have generally had poorer experiences with C19 relative to older generations and not just because the vaccination roll-out has been targeted in an inverse relationship with age.

Gender as an influence is statistically important in less than half of the models estimated, and generally, the impacts are small, i.e., males had a 6% higher probability of needing to self-isolate and were nearly 5% more likely to face a challenge with isolating and were (about 4%) more likely than women to believe the UK- and NHS-approved vaccines are very safe. Additionally, living in an area with greater levels of socioeconomic deprivation was not a statistically significant factor across many of the outcomes considered, although it was negatively correlated with attitudes on whether the UK government and NHS-approved vaccines are deemed very safe, i.e., moving from a IMD score of 2 to 80 reduced the probability of believing vaccines are very safe by around 11% to 12%. Being in a high deprivation area was also associated with a higher likelihood of facing challenges accessing COVID testing (8% higher moving from the lowest to highest deprivation scores).

Certain univariate results (reported in Section 3 of reference [15]) that showed that the non-White ethnic subgroup was less likely to have been vaccinated, were less likely to agree that vaccines were very safe, or were less likely to be able to make an informed decision on whether to vaccinate—all taken as indicators of vaccine hesitancy—were not confirmed by the multivariate model results. After controlling for other factors (principally age), the non-White subgroup only had a (statistically significant) higher probability of needing to test (nearly 13% higher), challenges in accessing testing (11% higher), and challenges getting vaccinated (12% higher relative to the White ethnic population). Thus, these results for the North East of England did not seem to indicate, ex post, that vaccine-hesitant attitudes are (cet. par.) more of an issue with the non-White ethnic population per se. The results presented here therefore do not seem to support the earlier analyses reported in references [3,5,7,11], but it is in accord with reference [9]. However, this was (at least in part) because of the inclusion of religious affiliation, which is considered next.

When the impact of religious affiliation is considered, and given that Muslims in particular almost all classified themselves as non-White, the results show that a higher vaccine hesitancy (associated with the safety of vaccines and making an informed decision about vaccination) is indeed prevalent in the North East of England but represents itself via its Muslim (and, to a lesser extent, Buddhist and Jewish) community. For example, those of a Buddhist faith were over 40% less likely to strongly agree that the UK government-approved vaccine is very safe (the result for NHS approval was weaker—a parameter estimate of −0.339 was only significant at the 12% level, while the result for making an informed decision on getting vaccinated was close to 0). As with Buddhists, those from the Jewish community were more sceptical of the safety of a UK government-approved vaccine (there is weaker evidence that they were also sceptical with regard to NHS approval or making an informed decision on getting vaccinated, with both estimates significant at the 11% level). Muslims were nearly 15% less likely to trust a UK government-approved vaccine (although this result was only significant at the 11% level) and even more hesitant about one approved by the NHS (they were 20% less likely to agree that the latter was very safe), and they were 20% less likely to strongly agree they could make an informed decision on getting vaccinated. Thus, taking together ethnicity and religious affiliation associated with ethnicity, there is robust evidence in favour of suggesting vaccine hesitancy is higher in the non-White population, as well as this subgroup having had a greater need to test for C19, and higher for the challenges associated with testing and getting a vaccine.

Those with a disability (especially the greater the incapacity) were more negatively affected by the pandemic. They were more likely to need to test for COVID and to self-isolate (especially those where the disability added more limits, who were 13% and 22% more likely to need a test and self-isolate, respectively). Disabled residents were (cet. par.) some 11–14% more likely to have received a vaccine, but they were more hesitant about endorsing the safety of the approved vaccines and had a lower probability of strongly agreeing they could make an informed decision about vaccination (for those with a greater disability, there was an 8% lower likelihood of strongly agreeing they could make an informed decision). The disabled faced more of a challenge with testing for C19, self-isolating, and getting a vaccine. The more disabled were, respectively, 11%, 24%, and 13% more likely to face such challenges.

The industry and occupation in which someone worked often mattered—e.g., in mining and quarrying, there were some large impacts: a 47% greater likelihood of needing to test, alongside a 32% higher probability of facing a challenge with testing, and a nearly 29% lower level of vaccination, together with a strong agreement (at around the 58–62% level) that vaccines were very safe. In other industry sectors, there were fewer impacts across the range of outcomes covered; some of the more striking results included a 31% lower likelihood of needing testing (cet. par.) for wholesale distribution, while those in transportation were 29% more likely to need testing. Notable occupation effects included a lower need to test or self-isolate for those in the admin and secretarial subgroup and greater challenges associated with testing for C19 for managers, professionals, skilled trades, and elementary occupations. Those less likely to be able to work from home in skilled trades; caring, leisure, and other service occupations; and process, plant, and machine operatives were also less likely to strongly agree that the approved vaccines were very safe. Compared to other occupation subgroups, (cet. par.), those in the caring, leisure, and other service subgroup were some 15% more likely to have received a vaccine.

When compared to the retired, the employed and unemployed/not economically active were less likely (cet. par.) to have received a vaccination by March 2021, and these groups were also more likely to show signs of vaccine hesitancy (i.e., they were less likely to agree that the approved vaccines were very safe). Those that experienced a change in their employment status during the pandemic were over 8% more likely to need testing and over 9% more likely to experience challenges when so doing. As to the impact of changes in household finances associated with the pandemic, for those experiencing declining incomes (poorer households), they were less likely to have been vaccinated and were less likely to strongly agree that the approved vaccines were very safe (including being some 5% less likely to strongly agree they could make an informed decision about being vaccinated). This group also faced more challenges with testing, self-isolating, and getting a vaccine. In contrast, households experiencing a positive effect on their finances (e.g., through lower outgoings leading to higher savings), all had higher levels of endorsement of the safety of the approved vaccines (as well as 7% stronger agreement that they could make an informed decision).

The sexual orientation of residents showed that gay and lesbian residents were some 12% more likely (cet. par.) to have received a vaccine, and they were also more likely to strongly agree that the approved vaccines were very safe. Those non-heterosexuals identifying as ‘other’ (not gay/lesbian/bisexual) were nearly 16% more likely to face a challenge in getting a vaccine. Household size had (cet. par.) few impacts; more adults were associated with a small (nearly 3%) increased challenge in self-isolating, while children in the household increased the need for testing by around 15.5%. Turning to household compositions, those with a couple and children, single parents, and those with adult children living with their parent(s) all had a lower probability of strongly agreeing that they could make an informed decision about the vaccinations (between 16% and 23% lower). In addition, those living with their parents were nearly 11% less likely to strongly agree that UK government-approved vaccines were very safe. Single-person households were (some 6%) less likely to need to test. Thus, overall, when compared to households comprising a couple (with no other residents), other types of households showed a greater propensity towards being vaccine-hesitant.

Lastly, there were some different outcomes depending on the local authority of the resident, e.g., those living in Darlington, Middlesbrough, and Redcar and Cleveland were more likely to need testing for COVID-19 (where the infection rates were relatively high), but it was only in Redcar and Cleveland that residents also had a greater challenge in testing and/or self-isolating. Middlesbrough had a nearly 12% higher likelihood of experiencing challenges linked to getting a vaccine.

## 4. Discussion

Given the major impact C19 has had on the economy, health, and the way people live their lives [19], tackling the pandemic continues to be a worldwide priority. At the time of writing, this means, in the UK, relying on the roll-out of vaccines to effectively immunise the population in order to reduce the transmission of the disease, hospitalisation levels, and deaths. The second major ‘plank’ in the ongoing control of COVID-19 is to ensure an effective means of testing and (self-)isolation of those infected, especially where new variants are concerned.

This paper makes a contribution by looking at the extent to which a wide range of personal characteristics and location are associated with the C19 questions asked in the North East Covid Survey to provide information relevant on how to run the most effective vaccination and ‘track-and-trace’ programmes. Issues such as the current approach to vaccination by priority groups (with precedence mostly age-related, with some inclusion of those deemed extremely vulnerable or, lower in the rankings, in particular, at-risk groups linked to prior medical conditions—see reference [20]) and whether this is optimal or needs amending to take account of other factors, such as ethnicity and/or location, are relevant policy questions needing examination. In this study, it was found that who needed to test or self-isolate, with their associated challenges, and those who received a vaccine and the challenges they faced, as well as which factors were the most associated with vaccine hesitancy, were not simply linked to the ages of the resident population. There are a range of other factors that are important, and the present study confirmed that ethnicity is important (especially when connected with religious affiliation), while another (linked to the more vulnerable and at risk) is disability. In contrast, the level of social deprivation of the area in which a resident lives seems less important (although whether the household is relatively poor is significant). Overall, these results showed that there are complex socioeconomic factors associated with the willingness to get a test, self-isolate, and the levels of vaccine hesitancy, such that, in the future, ensuring that (re-)vaccination and ‘track and trace’ programmes are successful may need to be better nuanced by references to such factors rather than adopting vaccination programmes that mostly just rely on age as the criteria for roll-outs. This also relates to the extent to which the government needs to combat health inequalities and, especially, the “anti-vaxxer” movement [21,22] through the better understanding of what makes certain people hesitant about undertaking C19 tests, self-isolating, and taking a vaccine. Hence, the results presented here lead to similar conclusions as in reference [11], who stated “…Our findings suggest, for example, that a “one size fits all” approach to mass media interventions represents, at best, a partial solution to increasing vaccination uptake and, at worst, a solution that backfires, amplifying existing inequalities. These findings suggest that future interventions need to be targeted to a range of sub-populations and diverse communities” (p. 6).

The major strengths of this study included its large sample size, representative of the population covered after weighting; the range of outcomes considered (rather than ex ante questions about the likelihood of whether residents are likely to get vaccinated); and the range of covariates (including religious faith, sexual orientation, and the industry/occupation of the respondent where relevant). A major caveat was the low level of representation of ethnic minorities in the North East region (some 4.6% of those aged 16+ years, when such minorities are classified as everyone except White British/Irish/Gypsy/Other White); moreover, when restricting the sample to those with full data on a range of characteristics (cf. Table 1), the weighted percentage for non-Whites fell to 3.1%. Thus, there is some evidence that ethnic minorities are relatively more reluctant to provide full information to this type of survey, and in any case, the North East (for this dimension) is not representative of other areas such as London, the Midlands, or even the North West [23].

## Figures and Tables

**Table 1 vaccines-09-00759-t001:** Weighted means and standard deviations of the explanatory variables.

Variable	Means	St. Dev.	Minimum	Maximum
Age of respondent	52.08	14.24	16	92
Male gender	0.48	0.50	0	1
Index of deprivation score in postcode lived (source: [18]) ^a^	22.84	15.75	1.53	78.01
White ethnic subgroup	0.97	0.17	0	1
Impact of C19 on household finances				
Negative impact on household finances	0.30	0.46	0	1
**No impact on household finances ^b^**	**0.48**	**0.50**	0	1
Positive impact on household finances	0.22	0.42	0	1
Religious affiliation				
Buddhist	0.00	0.05	0	1
Christian	0.53	0.50	0	1
Hindu	0.00	0.03	0	1
Jewish	0.00	0.05	0	1
Muslim	0.01	0.10	0	1
**No faith/religion**	**0.44**	**0.50**	0	1
Sikh	0.00	0.02	0	1
Other	0.02	0.14	0	1
Disability				
No	**0.73**	**0.45**	0	1
Yes, limited a little	0.20	0.40	0	1
Yes, limited a lot	0.07	0.26	0	1
Industry subgroups				
**Nonretired, n.a.**	**0.21**	**0.41**	0	1
Agriculture, Forestry and Fishing	0.00	0.05	0	1
Mining and Quarrying	0.00	0.03	0	1
Manufacturing	0.05	0.21	0	1
Electricity, Gas, Water supply	0.02	0.12	0	1
Construction	0.02	0.14	0	1
Wholesale distribution	0.01	0.07	0	1
Retail distribution	0.03	0.18	0	1
Transportation	0.01	0.12	0	1
Accommodation and Food Service Activities	0.02	0.12	0	1
Information and Communication	0.03	0.18	0	1
Financial and Insurance Activities	0.03	0.17	0	1
Professional, Scientific, and Technical	0.05	0.22	0	1
Administrative and Support Service Activities	0.05	0.22	0	1
Employment, Travel, Security	0.01	0.10	0	1
Public administration and defence; social security	0.08	0.27	0	1
Human Health and Social Work Activities	0.24	0.42	0	1
Arts, Entertainment and Recreation	0.03	0.16	0	1
Other Services	0.12	0.32	0	1
Occupation subgroups				
**None-retired, n.a.**	**0.20**	**0.40**	0	1
Managers, directors, senior officials	0.10	0.30	0	1
Professionals	0.27	0.45	0	1
Associate profession and technical	0.06	0.23	0	1
Admin and secretarial	0.16	0.37	0	1
Skilled trades	0.04	0.20	0	1
Caring, leisure, other services	0.08	0.27	0	1
Sales and customer services	0.05	0.21	0	1
Process, plant, and machine operatives	0.02	0.13	0	1
Elementary occupations	0.02	0.16	0	1
Economic status				
Full-time employed	0.44	0.50	0	1
Part-time employed	0.15	0.36	0	1
Unemployed/not active	0.16	0.37	0	1
**Retired**	**0.25**	**0.43**	0	1
Change in employment status during pandemic	0.24	0.43	0	1
Sexual orientation				
**Heterosexual**	**0.94**	**0.24**	**0**	**1**
Bisexual	0.02	0.13	0	1
Gay or Lesbian	0.03	0.19	0	1
Other (not straight/heterosexual)	0.01	0.08	0	1
Household size				
No. of adults	1.97	0.80	0	5
Children present in household	0.37	0.48	0	1
Household composition				
**Couple**	**0.41**	**0.49**	0	1
Couple with child/children	0.26	0.44	0	1
Other	0.02	0.15	0	1
Single parent	0.03	0.16	0	1
Single person	0.18	0.38	0	1
Single person or couple living with parents	0.03	0.18	0	1
Single person or couple with adult children	0.06	0.24	0	1
Local authority lived in				
**County Durham**	**0.26**	**0.44**	**0**	**1**
Darlington	0.05	0.23	0	1
Gateshead	0.11	0.31	0	1
Middlesbrough	0.05	0.22	0	1
Newcastle	0.13	0.34	0	1
N Tyneside	0.11	0.31	0	1
Redcar and Cleveland	0.07	0.25	0	1
S Tyneside	0.07	0.26	0	1
Sunderland	0.14	0.35	0	1
Observations	4533			

^a^ See Figure S1 in reference [15]. ^b^ Variables in bold are omitted as the benchmark subgroup in the estimated regression models.

**Table 2 vaccines-09-00759-t002:** (Weighted) marginal effects ($\partial \hat{p}/\partial x$) from the regression models.

Variable	Needed Testing ^a^	Needed Isolation ^a^	Had Vaccine ^a^	UK Govt-Approved Vaccine Very Safe ^b,c^	NHS Approved Vaccine Very Safe ^b,c^	Make Informed Vaccine Decision ^b,c^	Challenge Testing ^a^	Challenge Isolating ^a^	Challenge Getting Vaccine ^a^
Age of respondent	−0.003 ***	−0.002	0.013 ***	0.001	−0.001	0.001	−0.001 *	−0.002 *	−0.004 ***
Male	0.036	0.060 **	−0.001	0.037 *	0.039 *	−0.017	0.025	0.049 **	−0.007
Index of deprivation score in area	−0.000	0.001	−0.000	−0.001 **	−0.001 **	−0.000	0.001 *	0.000	0.001
White ethnic subgroup	−0.126 **	0.032	0.037	0.028	0.078	0.055	−0.107 **	−0.024	−0.119 **
Religious affiliation									
Buddhist	0.396 **	−0.008	−0.011	−0.413 **	−0.339	−0.012	0.092	0.052	0.103
Christian	0.068 ***	0.037 *	0.014	0.040 *	0.011	0.002	−0.016	0.026	0.001
Jewish	−0.061	0.035	−0.032	−0.360 **	−0.231	−0.212	−0.049	−0.054	0.005
Muslim	−0.088	0.106	0.119	−0.147	−0.203 **	−0.196 *	−0.151 *	−0.001	−0.068
Other	0.090	0.057	−0.183 ***	0.018	0.007	−0.032	0.007	0.028	0.018
Disability									
Yes, limited a little	0.057 **	0.140 ***	0.106 ***	−0.056 **	−0.059 **	−0.047 *	0.093 ***	0.135 ***	0.091 ***
Yes, limited a lot	0.131 ***	0.219 ***	0.139 ***	−0.085 **	−0.019	−0.084 **	0.110 ***	0.236 ***	0.126 ***
Industry subgroups									
Agriculture, Forestry and Fishing	−0.195	0.389 **	0.140	0.597 ***	0.451 **	0.492 **	−0.225	0.262 *	0.000
Mining and Quarrying	0.470 **	−0.192	−0.286 *	0.623 **	0.581 **	0.261	0.316 **	−0.345	0.000
Manufacturing	0.067	0.037	−0.022	0.069	−0.030	−0.134 **	−0.065	−0.043	−0.026
Electricity, Gas, Water supply	−0.003	0.107	−0.201 **	0.077	0.022	−0.063	−0.188	0.021	0.040
Construction	0.121	0.080	−0.059	0.068	−0.052	−0.078	−0.076	−0.022	−0.125
Wholesale distribution	−0.308 **	0.001	−0.050	0.315 **	0.003	0.077	−0.092	0.055	−0.093
Retail distribution	−0.046	0.103	−0.134 *	−0.051	−0.169 *	−0.079	−0.072	0.028	−0.083
Transportation	0.289 **	0.091	0.046	0.018	−0.125	−0.085	−0.027	−0.128	−0.214 **
Accommodation and Food Services	−0.047	−0.058	−0.014	0.042	−0.053	0.014	−0.096	−0.107	−0.005
Information and Communication	0.125	0.090	−0.062	−0.056	−0.072	−0.161 *	0.045	0.051	−0.066
Financial and Insurance Activities	0.019	0.134 *	−0.118 *	−0.013	−0.069	−0.111	−0.089	−0.029	0.012
Professional, Scientific, and technical	0.038	0.087	−0.112	−0.009	−0.014	−0.066	0.039	−0.014	0.011
Administrative and Support Service	0.072	0.117 *	0.010	−0.009	−0.078	−0.064	−0.033	0.033	0.032
Employment, Travel, Security	0.179	0.163	0.065	0.131	0.066	0.050	−0.166 *	0.079	−0.199 *
Public admin and defence, etc.	0.026	0.079	−0.095	0.052	−0.014	−0.015	−0.078	0.024	−0.043
Human Health and Social Work	0.005	0.081	0.061	0.050	0.011	−0.014	−0.025	0.038	−0.044
Arts, Entertainment, and Recreation	−0.051	0.024	−0.016	0.105	0.109 *	0.115*	−0.061	0.042	−0.044
Other services	0.006	0.100 *	−0.020	0.001	−0.042	−0.048	0.003	0.088	0.003
Occupation subgroups									
Managers, directors, senior officials	0.005	−0.009	−0.019	−0.006	0.020	0.048	0.082 *	−0.045	0.006
Professionals	0.004	−0.018	−0.013	−0.009	−0.009	0.044	0.080 *	−0.015	0.012
Associate profession and technical	−0.106	−0.121 *	−0.042	0.043	0.029	0.019	0.043	−0.058	0.059
Admin and secretarial	−0.101 *	−0.125 **	−0.030	−0.061	−0.041	−0.026	0.037	−0.078	0.024
Skilled trades	−0.028	0.035	−0.053	−0.179 **	−0.188 **	−0.080	0.165 **	0.117	0.048
Caring, leisure, other service	−0.012	−0.008	0.150 **	−0.094 **	−0.048	−0.078	0.081	−0.007	0.047
Sales and customer services	−0.058	0.054	0.008	−0.023	−0.017	0.022	0.090	0.054	0.002
Process, plant, and machine operatives	−0.072	−0.036	−0.059	−0.211 **	−0.196 **	0.033	0.096	0.070	0.003
Elementary occupations	0.119	0.037	0.014	0.018	−0.029	−0.071	0.119 *	−0.048	0.074
Economic status									
Full−time employed	0.061	−0.040	−0.133 ***	−0.090 **	−0.077 **	−0.013	−0.022	−0.035	0.021
Part−time employed	0.057	−0.057	−0.111 ***	−0.063 *	−0.061	−0.021	−0.008	−0.042	0.032
Unemployed/not active	−0.034	0.045	−0.149 ***	−0.120 ***	−0.117 **	−0.009	−0.084 **	0.047	0.056
Change in employment status	0.083 ***	0.037	−0.030	−0.026	−0.035	−0.025	0.093 ***	0.015	0.049 **
Negative impact on finances	−0.004	0.039	−0.049 **	−0.094 ***	−0.052 **	−0.051 **	0.036 *	0.099 ***	0.107 ***
Positive impact on finances	0.007	−0.014	−0.026	0.059 **	0.109 ***	0.065 ***	−0.023	−0.006	−0.002
Sexual orientation									
Bisexual	0.041	−0.058	0.012	−0.042	−0.060	−0.093	0.010	−0.016	0.031
Gay or Lesbian	−0.033	0.055	0.118 **	0.121 **	0.110 *	0.067	0.008	0.026	−0.092
Other (not straight/heterosexual)	0.012	−0.050	0.028	−0.088	−0.172	−0.086	0.099	−0.011	0.158 *
Household size									
No. of adults	0.006	−0.010	0.006	0.005	0.001	−0.005	0.006	0.026 *	−0.008
Children present in household	0.155 *	0.156	0.077	0.010	−0.049	0.086	0.096	0.091	0.103
Household composition									
Couple with child/children	−0.005	−0.019	−0.057	−0.054	−0.007	−0.161 *	−0.035	−0.017	−0.077
Other	0.096	0.169 **	0.034	0.091	0.050	−0.057	0.046	0.079	−0.141 **
Single parent	0.027	0.078	−0.136	−0.136	−0.109	−0.230 **	−0.031	0.131	−0.061
Single person	−0.058 *	−0.006	0.019	−0.042	−0.014	−0.031	−0.000	0.003	0.048
Single person/couple with parents	0.041	0.024	0.058	−0.106 *	−0.072	−0.026	0.057	0.003	0.078
Single person/couple with adult child	−0.028	−0.030	−0.035	−0.050	0.013	−0.206 **	−0.038	−0.054	−0.079
Local authority lived in									
Darlington	0.122 ***	−0.020	−0.027	0.042	0.012	0.029	0.006	−0.041	−0.023
Gateshead	0.042	0.043	−0.084 ***	0.007	−0.009	0.006	−0.023	−0.001	0.029
Middlesbrough	0.165 ***	0.052	0.070	0.004	0.024	0.026	0.045	0.055	0.118 **
Newcastle	0.039	−0.045	−0.050	0.067 *	0.074 *	0.062 *	0.034	−0.052	−0.035
N. Tyneside	0.014	−0.018	−0.047 *	−0.010	0.032	0.041	−0.042 *	−0.026	0.013
Redcar and Cleveland	0.088 *	−0.012	−0.031	−0.018	−0.036	−0.005	−0.068*	−0.090 **	0.060
S Tyneside	−0.033	0.075	0.092 **	0.015	−0.048	0.060	−0.036	0.012	0.058
Sunderland	0.034	0.055	−0.062 **	−0.004	−0.011	0.031	0.000	0.019	−0.004

Observations	4385	4395	4418	4435	4435	4435	4385	4395	4419
pseudo−R^2^	0.084	0.068	0.274	0.051	0.053	0.030	0.092	−0.017	0.092

*/**/*** represent significance at the 10%/5%/1% levels (based on robust standard errors). Table A1, Table A2, Table A3, Table A4 and Table A5 in Appendix A set out the actual survey questions asked for each model estimated. ^a^ Estimated using the probit regression model. ^b^ Estimated using the ordered probit regression model. ^c^ Only marginal effects for ‘strongly agreed’ reported.

## Data Availability

The SPSS and STATA code that cleaned the data and produced the results reported here are available on request. Those wanting to use the dataset should contact the Director of Public Health, Gateshead Council.

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
