# Peer review of "Experiences with Testing, Self-Isolation and Vaccination in North East England during the COVID Pandemic"

_vaccines, 2021, doi:10.3390/vaccines9070759_

Round 1

Reviewer 1 Report

Prof. Harris has extensively analyzed the huge dataset presented in this publication and  was able to make clear conclusions about the likelihood of North England inhabitants to need testing, to have access to testing, to trust vaccination recommendation, to have access to vaccination, and to get vaccinated against COVID19. This is overall a comprehensive study. I have mostly minor comments to improve the text and discussion:

  • In the abstract, do not use C19 abbreviation. Use COVID19.
  • Line 68: the bracket open was not closed.
  • The introduction should finish by a sentence explaining the aims of this publication. It will help the reader understand the purpose and the question(s) ask.
  • "BAME", "FT employed", and "PT employed" need to be defined.
  • Table 1: "impact of C19 on household finances" is presented twice within the table.
  • Why "heterosexual" is not used as a variable?
  • Table 2 is hard to read (i.e. it is hard to follow which line correspond to the text in the column 1). This table will benefit to draw a line in between each data.
  • Line 165: the author needs to remove "small" as 15 to 20% of the studied population is not a small percentage.
  • Line 187: "ceteris paribus" should be followed by "(cet. par.)" to define the following abbreviation.
  • One major comment I have, and that the author has partially addressed in the discussion, is about all the conclusions drawn for non-white, non-christian religious populations. There is only 3% of non white people in the dataset presented in this article and really few percentage of population that are not christians or not affiliated to any religion. Therefore, the conclusions made lines 188 to 190 and 235 to 243 appeared to be overstatements. Is this really representative of non-white and non-christian religious populations throughout UK? It appears to not be the case as the authors observes discrepancies with previous studies (see lines 240 to 242). Therefore, I will highly recommend that the author tunes down the statements made lines 188 to 190 and 235 to 243 as well as reinforce the statement made in the discussion about the limitation of the dataset presented here to be representative of non-white and non-christian religious populations throughout UK.
  • The non-binary gender are not taken into account in this study. Is there any studies that took non-binary gender into account? If yes, it will be interesting to discuss this data.
  • Line 301: the coma need to be removed after "vaccine".

Author Response

A response has been uploaded as a separate file; it contains the responses to all reviewers as changes made are not always specific to one reviewer.

Reviewer 2 Report

In the manuscript entitled, Experiences with Testing, Self-Isolation and Vaccination in 2 North East England during the COVID Pandemic, the author uses a population weighted sample of responses to a house-hold survey to identify associations between C19 testing, willingness to self-isolate, and vaccine hesitancy.   The rationale for the study is explained well in the introduction.  The study is also well designed.  My major concern with the study is the presentation of results.  It is very dense in content with many results.  The major points, sometimes get lost in the shear volume of information.  I recommend finding ways to emphasize or highlight major results.

Author Response

(The authors gave the same response as above.)

Reviewer 3 Report

Please define C19 in the Abstract.

Author Response

(The authors gave the same response as above.)

Reviewer 4 Report

This kind of work is very important in order to identify, in an epidemic or pandemic situation where testing and vaccination are possible, the groups that on the one hand have the highest need e.g. for testing, but on the other hand also have the highest hurdles in obtaining these necessary measures. In other words, how can these groups be given easy access to testing? Here, testing by company doctors (for certain professions) or by nursing staff (care of disabled persons) could be helpful, for example. In addition, the kind of multivariate analysis of the data that was carried out in this large-scale study offers the possibility to specifically identify the groups that are sceptical about vaccination, for example. Through this identification, one can then approach these groups directly and offer tailored information through the appropriate channels (e.g. through professional representatives, religious leaders or association bodies).
Also, for the problems mentioned, that lower income households have low vaccination rates is an important finding that authorities should use to come up with procedures to address this imbalance.

This paper provides an important foundation for the field, but in its form it could also serve as a blueprint for identifying the problem issues and population groups important to the pandemic situation. Particularly important is the use of multivariate analysis, which is not limited to one aspect, such as age, but relates the different aspects to each other.
However, I would like to make a few points that will make the study more appealing and easier to understand for the reader.
First, I would like to say that I find it unusual that the paper includes as a supplement a much more detailed version of the one submitted. I think it should rather be the case that the actual version contains all the important aspects and the supplements here should only be accompanying. For this reason, I would suggest the following. 
(1) the important aspects from the supplement, if not already done, should be included in the actual manuscript.
(2) the supplement is not a "second" paper, but is broken down into its various parts. For example, it can be referred to in the text if the reader wants additional information to be found in the supplement, in the manner of see supplement text 1.
(3) In the references, only the attached paper as a whole (reference 16) is referred to. It is not possible for the reader to form his or her own opinion, as he or she would have to work through the entire attached paper in order to find the statements made by the author. Instead, the author should refer to the respective appendices, be they texts, tables or figures. Each of these supplements should be referenced in the text itself, in the manner of "see Supplement Figure 1", "see Supplement Table 2", or "see Supplement Text 3".
These changes made the appended text not seem like a paper in its own right, but rather what it should be, an additional source of information that helps to correctly classify the statements, calculations and conclusions made, if one as a reader cares to do so.

In my opinion, such an important paper as this one should provide the authorities or persons responsible for managing an epidemic or, in this case, a pandemic, here related to a specific region, with the necessary information to master it as effectively as possible. The work should therefore reveal the problem areas. It should be defined which groups (determined by multivariate analysis), for example, do not want to be vaccinated because they are concerned and do not trust the public information, where there are problems with access to necessary testing, or where there are problems with self-isolation. And how to ensure an equitable vaccination strategy that works purely on medical grounds, regardless of income or occupation.
The information must be meaningful enough to enable those responsible to develop appropriate concepts and initiatives to specifically address the issues, as I have already described. In the case of vaccination scepticism, e.g. from certain religious groups, e.g. by involving the clergy etc. This information should not be too detailed for those responsible and also for the reader, but should be broken down into commonalities, e.g. with regard to contact with customers, the possibility of a home office or close contact with other workers, e.g. in the slaughterhouse or during field work (harvest). In my opinion, this enumeration of the many occupational groups is more confusing in its diversity and smallness than it is useful; here, less would be more, and the same applies to the occupation subgroups, which should rather be examined for its relevance with regard to the possibility of infecting third parties.
The data from the different administrative districts is difficult to understand for outsiders, here it would be good to show a map with the different districts in the paper (not with the sub-districts) and to mention the respective IMD score, for example, this would help the reader to better classify the data for the different districts. Also data on special features that represent an increased risk of infection or similar.

It would also be interesting to show the vaccination and testing rates for the different districts, as there may be differences between the districts that have an impact and influence the results.

Table 1 shows the group of people studied in a very differentiated way. This may be too differentiated for the reader, who may then get lost in details that are not relevant here. For example, the various occupational groups are listed in great detail. Here, for the sake of clarity, a shortening would be advisable, as already mentioned, the exact data could then be read in the supplements if of interest, but here, for example, the risk of infection (contact with customers or colleagues) would be of greater importance. What surprises me is that the level of education is not listed or is this equivalent to occupation or income in the study?
I wonder about the study of sexual preference. The data obtained from this (e.g. line 308) can then be interpreted in different ways, e.g. by discrediting/promoting or else by preference for a certain profession (e.g. in the healing professions). I find it difficult, as mentioned in the paper, to discuss the resulting points without further information.
The table extends over more than one page, so the headings of the columns should also be mentioned on the second and subsequent pages, otherwise you will always have to scroll back to the beginning of the table. The same applies to table 2. With regard to table two, the columns should really be placed one below the other and not offset, as is the case with the various districts, where the column "needed testing" is indented.

The important Table 2 is difficult to understand at first glance because it shows many data with different bases. For each heading, a different basis is used, with continuous variables and discrete ones. The discrete ones are based on the benchmark subgroups defined in Table 1.This point should be mentioned again in table two. Moreover, it is not always clear what the basis of the change is, e.g. in the case of age, is the value related to the difference of one year, i.e. from 20 to 21 or 40 to 41, or does it refer to the total time. There are many other things that remain unclear, which should be better clarified here. 

I did not find other statements in the table. For example, there are statements about non-white persons in the text (line 234 ff), which I do not find in the table, because only white persons are mentioned there. I ask that this be clarified. And what are the 11% (line 256) or 12% level (line 251) the author is talking about?

A few questions remain open. Does a prioritisation of vaccination, e.g. according to age, have an influence on the values found with regard to the proportion of vaccinated persons among the younger ones (working population)? Do some groups have problems understanding the language, so that certain information did not reach them at all or even this survey was not conducted correctly? How can something like this be ruled out, or how do those responsible for controlling the crisis have to act in order to reach such people?

The author himself points out the limitations regarding minorities, which represent a slight restriction of the significance for these groups (only approx. 140 of 4500), but this should not diminish the fundamental significance of the paper, one just has to keep this in mind when interpreting it. But the tools used can still help in studies with a larger cohort of minorities to make the necessary statements.

All in all: Reduce the tables to the most important findings, the rest can be shown in the supplement. Summarise these findings in an easily understandable graphic, if possible, and design the attached text as a real supplement collection.

Author Response

(The authors gave the same response as above.)

Round 2

Reviewer 4 Report

Thank you very much for the clarifying comments. I can now endorse the publication in its present form.